# Characterizing and Removing Artifacts Using Dual-Layer EEG during Table Tennis

**DOI:** 10.3390/s22155867

**Published:** 2022-08-05

**Authors:** Amanda Studnicki, Ryan J. Downey, Daniel P. Ferris

**Affiliations:** J. Crayton Pruitt Family Department of Biomedical Engineering, University of Florida, Gainesville, FL 32611, USA

**Keywords:** electroencephalography, dual-layer, motion artifact, table tennis

## Abstract

Researchers can improve the ecological validity of brain research by studying humans moving in real-world settings. Recent work shows that dual-layer EEG can improve the fidelity of electrocortical recordings during gait, but it is unclear whether these positive results extrapolate to non-locomotor paradigms. For our study, we recorded brain activity with dual-layer EEG while participants played table tennis, a whole-body, responsive sport that could help investigate visuomotor feedback, object interception, and performance monitoring. We characterized artifacts with time-frequency analyses and correlated scalp and reference noise data to determine how well different sensors captured artifacts. As expected, individual scalp channels correlated more with noise-matched channel time series than with head and body acceleration. We then compared artifact removal methods with and without the use of the dual-layer noise electrodes. Independent Component Analysis separated channels into components, and we counted the number of high-quality brain components based on the fit of a dipole model and using an automated labeling algorithm. We found that using noise electrodes for data processing provided cleaner brain components. These results advance technological approaches for recording high fidelity brain dynamics in human behaviors requiring whole body movement, which will be useful for brain science research.

## 1. Introduction

Electroencephalography (EEG) facilitates mobile brain–body imaging (MoBI) because of its high temporal resolution, light weight, and easy portability [1,2]. Traditionally, EEG studies have been constrained to stationary tasks due to physiological and non-physiological artifacts associated with movement. Recent advances in signal processing approaches, such as independent component analysis [3,4,5], artifact subspace reconstruction [6], and a variety of other techniques [7], and hardware approaches, such as active electrodes [8] and bundling cables [9], allow for high fidelity EEG recordings during whole body movements [10,11]. However, there is still a limit to the magnitude and types of movements a mobile EEG experiment can conduct without sacrificing its signal-to-noise ratio [12,13,14]. 

Prior knowledge of artifacts (i.e., reference signals) may aid in signal processing. Separate electro-oculogram (EOG) and neck electromyography (EMG) channels help remove ocular artifacts [15,16,17] and myogenic artifacts [18] in EEG data. Head movement, measured with inertial measurement units or motion capture, can be used to get rid of motion artifacts as well. Some approaches perform independent component analysis (ICA) and reject components that correlate with head movement above some threshold [19,20]. Other approaches use adaptive filtering to remove noise from the EEG data that correlates with head movement [21]. Yet, Kline et al. [14] found that head acceleration during treadmill walking did not correlate well with movement artifacts recorded by electrically isolated EEG channels on the head. They conjectured that head acceleration may not record the same movement artifacts that EEG electrodes record. A follow-up study by Symeonidou et al. found that cable sway actually contributed much more to EEG motion artifacts than scalp-electrode perturbations due to electrode mass and surface area [9].

In the dual-layer EEG approach, noise channels provide an alternative representation of motion artifact [22,23,24]. Inverted noise channels are electrically isolated and mechanically coupled to scalp channels with their wires wrapped together with tape. Theoretically, noise channels should have a high correlation with the non-physiological artifacts on scalp channels (i.e., cable movement, electro-magnetic interference, electrode movement). Noise channels may paint a more complete picture of the cable-induced artifact than other sensors. The dual-layer EEG approach was first introduced and validated in a series of phantom head studies [22,23,24]. Nordin et al. demonstrated that signal quality improved after noise cancellation with dual-layer electrodes compared to a single layer of electrodes for a phantom head moving vertically on a custom motion platform [24]. In human data, the dual-layer approach has been applied to studies with participants walking on a treadmill [25] and navigating obstacles [23]. While the dual-layer approach has proven advantages in some situations, further investigation into why it works would benefit the field. A time series correlation, like in Kline et al. [14], may help us understand how noise channels relate to scalp channels.

Most studies characterizing or investigating motion artifacts focus on locomotion [13,26,27]. Often, harmonics of the stepping frequency are indicative of noise contamination [12,13], and some artifact removal strategies rely on a cyclical, repetitive pattern [26]. However, not all mobile paradigms follow a cyclical path [28,29]. Noise from primarily lower limb, sagittal plane movements in walking may also differ from tasks relying on whole-body, multi-plane movements. There is a need to investigate motion artifacts in paradigms other than treadmill locomotion or simulated locomotion.

Table tennis is a useful sport to study because it is a whole-body, responsive, goal-directed activity that requires upper and lower limb coordination [30]. Many neuroscience studies could use table tennis as a testbed to investigate visuomotor feedback, anticipation, strategic decision-making, object interception, performance monitoring, and more. While previous studies have looked at functional brain data in simulated or virtual racquet sports [31,32,33,34,35], it would be useful to study the neural correlates of live table tennis considering the importance of ecologically valid conditions [36,37,38,39,40,41,42,43]. Because the human brain integrates multi-modal sensory and motor stimuli under dynamic and complex conditions, it is important to study the brain’s response in similar conditions.

Properly handling EEG artifacts during sport is important for confidence in interpreting neural correlates of motor behavior. Players must turn their head and body to hit the ball, and the rotational movements may induce a different cable artifact than locomotion. Depending on the consistency of play, motions may be considered cyclical or non-cyclical and may require different artifact removal strategies. To improve the fidelity of the neural information and interpretation of brain functions, we must reduce the impact of motion artifacts on recorded EEG data.

The purpose of this study was to characterize and identify strategies to remove artifacts in EEG data for a discrete, responsive, whole-body task such as table tennis. We compared individual scalp channels, noise-matched channels, neck muscle channels, and accelerometer data from the head and body to determine the efficacy of different reference signals in capturing noise. Participants played in an assortment of table tennis drills and games. We hypothesized that individual channel-level movement artifacts recorded by noise electrodes would correlate with the raw electrocortical data recorded by scalp electrodes. We also sought to identify processing strategies for cleaning table tennis data. A recently introduced dual-layer approach using canonical correlation analysis on the scalp and noise electrodes to find and reject correlated components, termed iCanClean [44], was benchmarked against single-layer processing approaches.

## 2. Materials and Methods

### 2.1. Participants

We collected data from 37 participants (ages 23.5 ± 6.7 years, mean ± SD; 13 females) with no self-reported musculoskeletal or neurological injuries and with normal or corrected-to-normal vision. Participants had a wide range of table tennis and racquet sport experience. All participants self-identified as right-hand dominant. Our exclusion criteria were an inability to follow directions or make contact with the ball. Participants gave written informed consent, and the University of Florida Institutional Review Board approved our protocol.

### 2.2. Dual-Layer Approach

Participants wore a custom-made dual-layer EEG system (ActiCAP snap sensors; BrainVision LLC., Morrisville, NC, USA) [45] made up of 120 scalp electrodes and 120 noise electrodes (Figure 1a).

Scalp electrodes were mechanically joined to inverted noise electrodes using 3D-printed couplers, and wires were secured using tape, so both cables experienced similar motion [9]. As described in [24], the scalp electrodes recorded a mixture of biological signals and motion artifacts, whereas the electrically isolated noise electrodes recorded motion and non-biological artifacts. Conductive fabric (EeonTex elastic piezoresistive fabric, LTT-PI-100; Marktek Inc., Chesterfield, MO, USA) acted as an artificial skin circuit and bridged the noise electrodes. We re-purposed eight of the original 128 scalp electrodes (TP9, P9, PO9, O9, O10, PO10, P10, and TP10) to measure neck muscle activity (left/right and upper/lower sternocleidomastoid and trapezius). During participant set-up, we verified that electrode impedance values were below 20 kΩ.

Four LiveAmp 64 amplifiers (BrainVision LLC., Morrisville, NC, USA) logged the EEG data at 500 Hz. Each LiveAmp recorded data from a single layer of electrodes from one hemisphere. The online reference and ground electrodes were located at CPz and Fpz, respectively, and were shared across hemispheres. However, the reference and ground electrodes for the scalp and noise layers were kept separate (electrically isolated). We shortened the BrainVision ribbon cables to 14 cm and housed the scalp and noise-matched amplifiers in two 3D-printed cases. A small backpack (35 × 25 × 11 cm) filled with lightweight foam kept the amplifiers secure (Figure 1b). Straps were adjusted so the backpack rested on the upper back of each participant. The total weight of the system was 2.7 kg (6 lbs).

### 2.3. Inertial Measurement Units

We placed inertial measurement units (WaveTrack Inertial System, 2000 Hz; Cometa Systems, Bareggio, Italy) on the handles of two wooden paddles, the ball machine (Robo-Pong 2040+ Ping Pong Robot; Newgy Industries, Inc., Hendersonville, TN, USA), table tennis table (Joola, 15 cm thick), and net to measure peaks in acceleration for timing of events. For the last twenty participants we placed an inertial measurement unit (IMU) on one amplifier inside the backpack to measure linear and angular motion of the participant’s body. This IMU’s acceleration also acted as a check for time synchronization since we could compare it to the EEG amplifier’s built-in accelerometer. For the last four participants, an IMU was placed on the participant’s forehead with double-sided tape since head motion in mobile EEG studies is thought to be a common source of noise [19,46,47]. For the last two participants, we positioned an IMU on the participant’s lower back underneath the backpack, secured with medical dressing, to confirm that the IMU on the amplifier inside the backpack had similar motion to the participant’s body motion. Since only a subset of participants had the IMU on the amplifier inside the backpack, we used the last twenty participants for further analysis in this study. Each sensor’s memory logged the IMU data. Pulses from an Arduino timer module every five seconds synchronized the EEG and IMU systems.

### 2.4. Experimental Protocol

Participants played table tennis with a ball machine and a human player (Figure 1c,d, Appendix A), broken into four 15-min blocks of play. Breaks were given between blocks and as needed. In a single block, participants played cooperatively or competitively with a human player for a continuous 7.5 min and played with the ball machine for three short back-to-back trials, 2.5 min each. Thus, there were 16 trials for a given experiment, four trials in each block. For cooperative trials, we instructed participants to work together with the human player to keep the ball in play. For competitive trials, participants were instructed to try winning 21-point games against the human player, switching serves every five points. The human player was experienced enough to scale their play to match the level of each participant.

Within a block, the ball machine trials alternated between stationary and moving conditions. For the stationary condition, the ball was fed to the center of the table at a rate of approximately once every two seconds (0.5 Hz). The ball’s trajectory was predictable, and the participant did not have to move their feet to intercept the ball. For the moving condition, the ball machine oscillated and fed balls toward all directions on the table at the same rate of 0.5 Hz. Half of the ball machine trials fed the ball to the participant on one bounce and the other half fed the ball on two bounces, replicating a serve. The angle and speed of the feed were adjusted so that the ball landed halfway between the net and the end of the table. The order of trials among and within blocks was randomized. Participants were instructed to avoid clenching their jaw and to stay as relaxed as possible. We did not instruct participants to hit a specific target, nor did we constrain participants to use a specific type of shot.

We collected a five-minute standing baseline at the beginning and end of all the table tennis trials. Participants stood comfortably while holding the paddle at their side and fixating their eyes on the ball machine. The total amount of data collected, excluding breaks, was 70 min.

### 2.5. Artifact Characterization

#### 2.5.1. Preprocessing

We processed data using custom MATLAB scripts (R2020A) and EEGLAB functions (v2021.0) [48]. Scalp, noise, and muscle channels were 1 Hz high-pass filtered with a zero-phase, finite impulse response (FIR) filter using the eegfiltnew function to remove drift. Scalp, noise, muscle, and IMU data were aligned and merged into a single dataset using the TTL synchronization pulses from the Arduino timer module. We marked hit events when the first derivative of the resultant acceleration of the ball machine or paddle IMUs exceeded 0.75 gravity. We visually inspected and manually separated trials into individual datasets based on continuous blocks of hitting events. For each trial, we down-sampled the data to 250 Hz and rejected bad channels that were outside of three standard deviations away from the median of all other channels for scalp and noise electrodes separately. We cropped trials to the first and last hit events with 2 s of padding on each side. The IMU channels were 1 Hz high-pass filtered using eegfiltnew to remove drift, and we combined the *X*, *Y*, *Z* axes by taking the root-mean-square. The noise channels were 50 Hz low-pass filtered to remove significant line noise contamination that only appeared on noise channels. Because we repurposed eight scalp electrodes to record neck muscle activity, the raw neck muscle data shared a common reference with scalp data (CPz). We subsequently took a weighted difference (i.e., bipolar measure) between superior and inferior muscle channels at each muscle site: left and right sternocleidomastoid (SCM) and left and right trapezius. Taking the difference removed the shared reference channel.

We compared power spectral densities, event-related spectral perturbations, and time series correlations of reference noise sensors and scalp electrodes to characterize artifacts (Figure 2).

#### 2.5.2. Power Spectral Density

We computed power spectral density plots for each participant. Data were epoched −1 to 1 s around the participant hit event. For the standing baseline trials that did not have events, we marked pseudo-hit events every 2 s. We concatenated epochs from the same condition and calculated the power spectral density of each scalp and noise-matched electrode, bipolar neck EMG channel, and head and body (i.e., backpack) IMU channels using the spectopo function (pwelch method with non-overlapping hamming windows, 250 window length). The power spectral densities were then averaged across participants.

#### 2.5.3. Event-Related Spectral Perturbation

Time-frequency analysis allowed us to assess the frequency content across an average “swing cycle”. We defined a “swing cycle” as the anticipation, swing, and recovery associated with three events: (1) when the ball was first presented to the participant, either as a ball machine feed or hit by another human player; (2) when the participant hit the ball; and (3) when the ball was presented to the participant for the next hit. To capture all three events, we epoched each condition’s data −1 to 3.5 s around the participant hit event and concatenated epochs from the same condition. Then, we used the newtimef function to perform time-frequency analysis with Morlet wavelets whose cycles increased linearly with frequency (3 cycles at the lowest frequency and 64 cycles at the highest frequency). Within the newtimef function, we specified a “padratio” of four and epochs were time-warped to the median latencies of hit events across all conditions. The average power across all time points in the epoch for the standing baseline condition was the spectral baseline removed from the time-frequency points in the other hitting conditions. To find statistically significant time-frequency pixels, we bootstrapped individual participant’s event-related spectral perturbations with 200 exemplars at a significance level of alpha 0.05 using the bootstat function.

#### 2.5.4. Correlation Analyses

To find correlations, we took the absolute value of the correlation between epoched scalp electrode data (−1 to 1 s around the participant hit event) and noise-matched electrodes, muscle channels, and resultant acceleration of the head and body IMUs. Correlations were computed within the standing baseline and for each condition: stationary hitting and moving hitting with the ball machine, and cooperative and competitive playing with a human player. For the correlation between scalp and muscle channels, we used a 3 Hz high-pass filter (eegfiltnew) to get rid of low-frequency motion artifact and to focus on the muscle activity. To find group average correlations, Pearson’s R correlation coefficients were converted to the Fisher’s Z transform, averaged across epochs, averaged across participants, and back-calculated to Pearson’s R correlation coefficients [49].

### 2.6. Artifact Removal

#### 2.6.1. Artifact Removal Strategies

We employed four different pipelines to compare artifact removal strategies (Figure 3). The Minimal pipeline had the same cleaning as described in Section 2.5.1 “Preprocessing” with a couple additional steps. Rejected channels were spherically interpolated using default electrode locations, and scalp and noise channels were separately re-referenced to their averages with zero loss of rank [50]. The Minimal & Time Reject pipeline used a part of the artifact subspace reconstruction algorithm [51] to reject noisy time windows after the Minimal pipeline was applied. We set a conservative standard deviation threshold to 30 [52] and window criterion to 0.3. The exploratory Minimal, 120iCan & Time Reject pipeline [44] used iCanClean after the Minimal pipeline and with subsequent time window rejection using the artifact subspace reconstruction algorithm. For iCanClean, we applied canonical correlation analysis in a 2-s sliding window to find subspaces of the 120 scalp electrodes that were strongly correlated with subspaces of the 120 noise electrodes. The subspaces with the strongest correlation (r^2^ > 0.85) were rejected. Because we did not know how many noise electrodes were needed for the dual-layer approach, we also ran the iCanClean pipeline after down-sampling to 64, 32, 9, 5, and 1 noise electrodes that were evenly spaced across the head. According to a phantom head study [22], as few as 32 noise channels were effective at capturing and removing motion artifacts during simulated human walking. Table tennis could arguably be a more dynamic and complex task, so more noise electrodes may be needed for effective cleaning. Lastly, since most motion artifacts occur at low frequencies [14], we employed a Minimal, 3 Hz & Time Reject pipeline to filter the scalp electrodes at a 3 Hz cut-off value after the Minimal pipeline and with subsequent time window rejection. The 3 Hz high-pass filter substituted the exploratory iCanClean processing step.

#### 2.6.2. Pipeline Comparison

Scalp channel data from the hitting conditions were selected and concatenated together per participant. We ran principal component analysis on the data to ensure rank by reducing the principal components by the maximum number of scalp channels that were interpolated for a given participant across trials. Next, we used an adaptive mixture independent component analysis (AMICA) [53,54], which has been shown to outperform other methods of independent component analysis [55]. The weight matrix from each pipeline after AMICA was applied to the dataset from the Minimal pipeline to ensure a fair comparison. We fit each independent component to an equivalent dipole using DIPFIT3.3 [56] with default electrode locations and a three-layer boundary element model of the default Montreal Neurological Institute (MNI) brain template. To objectively measure the quality of the decomposition, we computed a dipolarity metric as the number of components with residual variance less than 15% [55]. We also computed the number of brain components that the ICLabel plug-in [57] classified with a brain probability of greater or equal to 75%. In a supplementary analysis, we computed the number of components that ICLabel classified as muscle, eye, heart, line noise, channel noise, and others with a probability greater or equal to 75%.

To statistically test differences in processing pipelines, we performed a one-way repeated measures ANOVA (alpha = 0.05) on the dipolarity metric and ICLabel metric for each pipeline using the MATLAB (R2020A) function fitrm where the subject was the predictor variable, each metric was the response variable, and the pipeline was the within-subjects variable. To account for Type I error, we adjusted our results using the false discovery rate correction with the Benjamini and Hochberg method [58].

## 3. Results

### 3.1. Artifact Characterization

The shapes of the power spectral density plots varied by sensor type (Figure 4). The noise electrodes exhibited higher power in low frequencies and their slope was steeper than the other sensors. Scalp electrodes followed a common 1/f curve [58] but had an increase in power in the alpha and beta frequency range, most noticeable in the standing baseline condition.

The mobile hitting conditions (i.e., hitting and competitive) had more power in higher frequencies than the standing baseline, stationary, and cooperative conditions. The resultant body IMU acceleration exhibited a much bigger change in low-frequency power between the standing baseline and all other hitting conditions. The body IMU had more power in frequencies 5–30 Hz than the noise electrodes. The bipolar neck EMG channels had low power in low frequencies and high power in high frequencies with a broadband increase in power for the mobile hitting conditions.

All sensor types and hitting conditions showed statistically significant time-frequency pixel changes in power compared to the standing baseline condition (Figure 5).

The noise electrodes’ change in power was consistent across electrode locations and were concentrated in low frequencies around the hit event (vertical dashed line). The increase in power was more diffuse across the swing cycle in the moving and competitive than stationary and cooperative conditions. The scalp electrodes show a similar pattern of spectral power increase around the hit event that is more diffuse in the moving and competitive conditions. However, FCz and C5 scalp electrodes also exhibit alpha desynchronization relative to the standing baseline that was not present on any reference noise sensor. Scalp electrodes showed statistically significant increases in power in high frequencies that were more pronounced in the moving and competitive conditions, especially for the POz scalp electrode. The body IMU had similar time-frequency characteristics as the noise electrodes but showed more broadband changes across the swing cycle. As expected, most of the power in the neck EMG channels were in high frequencies, concentrated around the participant hit event. For non-significance masked time-frequency plots, refer to Appendix A.

The Pearson R correlation between individual scalp channels and the other sensors varied by electrode location (Figure 6). Muscle channels correlated most with scalp channels at the back and sides of the head. The left trapezius and right sternocleidomastoid channels correlated with the scalp channels more than the right trapezius and left sternocleidomastoid (Appendix A). There were no apparent trends in laterality between left and right muscles.

The correlation between the individual scalp and noise-matched electrodes increased in the hitting conditions compared to the standing baseline condition. There was a slightly higher correlation between the scalp and noise-matched electrodes at the front of the head during the standing baseline condition and on top of the head during the hitting conditions. We observed little difference in correlation between the scalp and noise-matched electrodes across the four hitting conditions.

The scalp electrodes correlated less with the body and head IMUs than the noise-matched electrodes. The body and head IMU correlation with the scalp electrodes showed a similar increase in the four hitting conditions compared to the standing baseline condition. Again, there was not much difference in correlation between the four hitting conditions. The cooperative condition may show a slight decrease in correlation between the scalp electrodes and body and head IMU. Since only three participants had head IMU data, refer to Appendix A for spectral features of the head and body IMU for a small subset of participants.

### 3.2. Pipeline Comparison

Overall, the Minimal, 120iCan & Time Reject pipeline outperformed the other cleaning pipelines (Figure 7 and Figure 8). On average, the Minimal, 120iCan & Time Reject pipeline resulted in 15 more components that passed the dipolarity threshold of <15% residual variance compared to the Minimal and Minimal & Time Reject pipelines (Figure 7).

The Minimal, 3 Hz & Time Reject pipeline had a similar number of dipolar components, although slightly less than the Minimal, 120iCan & Time Reject pipeline. The ICLabel metric showed a similar trend (Figure 8).

There were around six or seven components that were labeled as >75% brain by ICLabel after the Minimal and Minimal & Time Reject pipelines. The Minimal, 120iCan & Time Reject pipeline doubled that outcome, returning around 12 components that were labeled as >75% brain. The Minimal, 3 Hz & Time Reject pipeline had around 10 components on average that were labeled as brain. ICLabel classified fewer components as “other” for the Minimal, 120iCan & Time Reject and Minimal, 3 Hz & Time Reject pipeline. Performance gradually decreased, shown in both the Dipolarity and ICLabel metrics (Appendix A), as the number of noise electrodes used in iCanClean decreased. Refer to Appendix A for dipolar component properties of each pipeline. All pipelines had a significantly different number of dipolar components at the alpha 0.05 significance level. The Minimal and Minimal & Time Reject pipelines had significantly fewer dipolar components than the Minimal, 3 Hz & Time Reject and Minimal, 120iCan & Time Reject pipelines at the 0.01 significance level. We found the same trends for the ICLabel metric, except the Minimal, 3 Hz & Time Reject pipeline showed no difference from the Minimal, 120iCan & Time Reject pipeline at the 0.05 significance level.

## 4. Discussion

### 4.1. Artifact Characterization

Our results mildly support the hypothesis that individual noise electrodes correlate with minimally processed electrocortical data on the scalp channels. The time series correlations between scalp and noise electrodes were smaller during standing baseline than any hitting condition, indicating that both sets of electrodes were experiencing similar movement artifacts. However, we did not observe increases in correlation for more mobile hitting conditions (moving > stationary and competitive > cooperative). We expected to see differences between these conditions, similar to the spectral data (Figure 4 and Figure 5).

It is possible that our table tennis conditions were not dynamic enough to produce large motion artifacts. There was a clear difference in magnitude of the noise electrodes across the conditions (Figure 2), but other tasks such as tennis or volleyball would likely have even greater magnitudes. It would be interesting to compare our magnitude of movements and noise channel data to a wide range of motor behaviors. Lastly, our participant preparation may have eliminated a great deal of noise from the recordings. Two layers of electrodes are like using a secondary stabilizing cap that is thought to help with EEG data quality [21]. We shortened the ribbon cables to reduce as much cable-induced artifact as possible [9], but it is difficult to determine how much this affected the scalp channel’s motion artifact.

There may be better ways to relate scalp channels and reference noise signals. Kilicarlson et al. [21] used a Volterra-based non-linear mapping to characterize the motion artifact in EEG data with head acceleration. A non-linear representation may better encompass the effect of noise on the scalp sensors than a linear correlation.

The overlap in spectral features between the scalp and reference noise sensors provides insight for future analyses. For this data, we must be careful to interpret neural data in the delta and theta range (<7 Hz) which may have motion artifact contamination. Noise electrodes showed increases in power relative to standing baseline at low frequencies, which increased in magnitude with more mobile hitting conditions (Figure 5). The theta power increase around the participant hit event for scalp electrodes was similar but not identical to the noise-matched electrodes across conditions. The scalp electrodes at the back and sides of the head (POz and C5) shared similar spectral features with the neck muscle channels. We can attribute the increase in power at the high frequencies to muscle contamination. However, because we only observed alpha desynchronization on the scalp electrodes (Figure 4 and Figure 5), we can be more confident that these are neural sources. A comparison of spectral features before and after each step of the pipeline may be helpful in determining which processing steps are best.

### 4.2. Pipeline Comparison

iCanClean helped clean the data, as evidenced by the dipolarity and ICLabel metric (Figure 8). However, more work is needed to tune the algorithm’s parameters, such as the window length and component correlation threshold for removal. The second-best cleaning approach was the Minimal, 3 Hz & Time Reject pipeline, which replaced iCanClean with a 3 Hz high-pass filter. This finding aligns with other literature showing that higher filter cut-offs may be necessary for mobile EEG data [59]. However, in theory, the iCanClean approach may better preserve electrocortical data in lower frequency bands (<5 Hz). Additionally, iCanClean could use other reference signals such as neck EMG to clean high-frequency myogenic artifacts from scalp channels. Future work should investigate the capabilities and application of iCanClean.

There was a gradual decline in the quality of decomposition for decreasing the number of noise electrodes used in iCanClean (Appendix A). In a practical application of the dual-layer approach, 64 noise channels may be sufficient to represent the cable-induced artifact. In the future, a principal component analysis may be helpful to see how many noise electrodes are needed to represent most of the variability in the data. A pipeline comparison with a full parameter sweep is beyond the scope of this work. In this study, we presented an analysis of a few different processing strategies as an exploratory step that provides insight for prospective analyses. Future work ought to test different dual-layer processing methods and techniques that use reference signals for cleaning.

Another approach to compare the data quality for different processing pipelines would be to use a ground truth signal from non-human studies. Electrical head phantoms allow a direct comparison of input and output signals to test different signal processing approaches [9,22,23,24]. In human data, dual-task event-related potentials can estimate the signal-to-noise ratio for different processing approaches in the absence of ground-truth data. In this study, we evaluated different pipelines using dipolarity and ICLabel metrics after an ICA decomposition [55]. We plan to analyze our brain data in component-space in future publications, so we sought a pipeline that gave the best component decomposition. Dipolarity tests the assumption that sources are spatially fixed and temporally distinct. We used ICLabel as an estimation of the number of brain components separated after ICA as it is becoming commonly used for that purpose [57]. However, the ICLabel plug-in was primarily trained on stationary behavioral tasks rather than whole body motor tasks like table tennis. The inclusion of a wider range of motor behaviors into the ICLabel training set in the future would expand its capabilities.

## 5. Conclusions

Understanding the neural correlates of natural movement under ecologically valid conditions is important [2,37,38]. This study advances techniques for mobile brain imaging by analyzing artifacts in a whole-body, responsive task. Table tennis is a useful sport to investigate visuomotor feedback, anticipation, strategic decision-making, object interception, performance monitoring, and more. Many studies use simulated or virtual racquet sport paradigms [31,32,33,34,35], while very few studies measure brain processes during active play [60]. Properly handling artifacts in mobile table tennis experiments is important for confidence in interpreting neural data in the sport. We found that the dual-layer approach does well to characterize motion artifacts affecting scalp channels during a dynamic sport like table tennis. The dual-layer sensor approach combined with iCanClean may also have use in sensor processing applications outside of EEG and brain research.

## Figures and Tables

**Figure 1 sensors-22-05867-f001:**
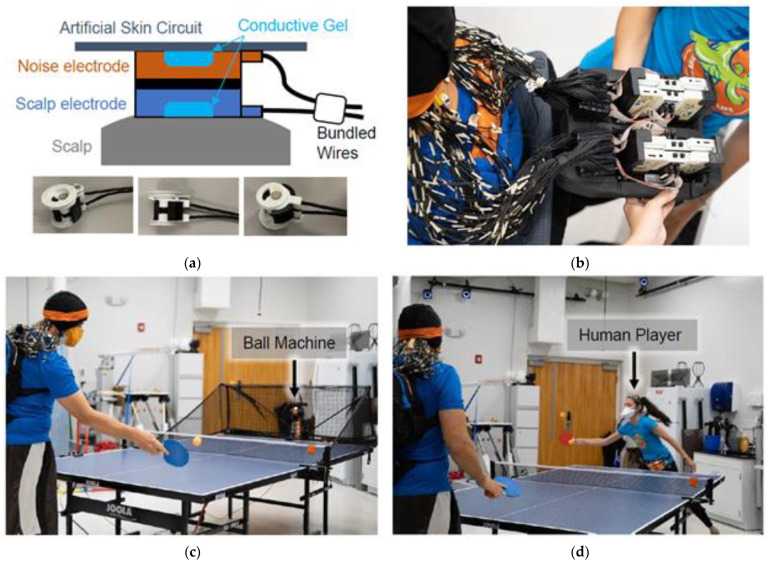
Table tennis experiment. (**a**) Dual-layer EEG approach. (**b**) The custom backpack insert for BrainVision amplifiers. (**c**) Stationary and moving hitting conditions with a ball machine. (**d**) Cooperative and competitive conditions with a human player.

**Figure 2 sensors-22-05867-f002:**
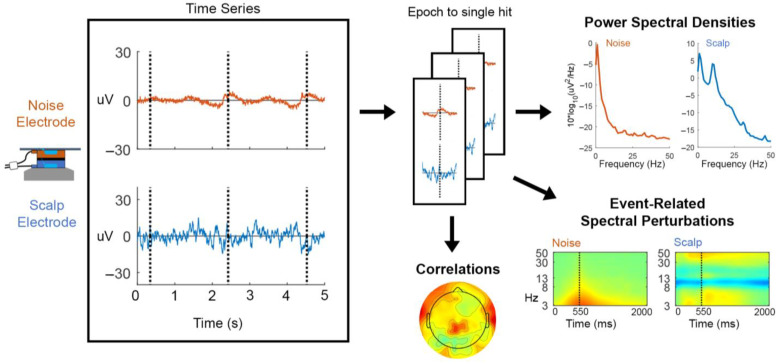
Overview of the artifact characterization process. Time series data of all dual-layer electrodes (noise electrode time series in orange and scalp electrode time series in blue) were epoched to single participant hit events. We computed (1) power spectral densities of each participant’s sensor and for each condition (standing baseline, stationary, moving, cooperative, and competitive), (2) event-related spectral perturbations which were time-warped to an average “swing cycle” whose baseline was the average power from the standing baseline condition, and (3) linear time series correlations between all scalp and noise-matched electrodes. This process was repeated for all reference noise sensors, including the bipolar neck EMG, body, and head IMU acceleration data.

**Figure 3 sensors-22-05867-f003:**
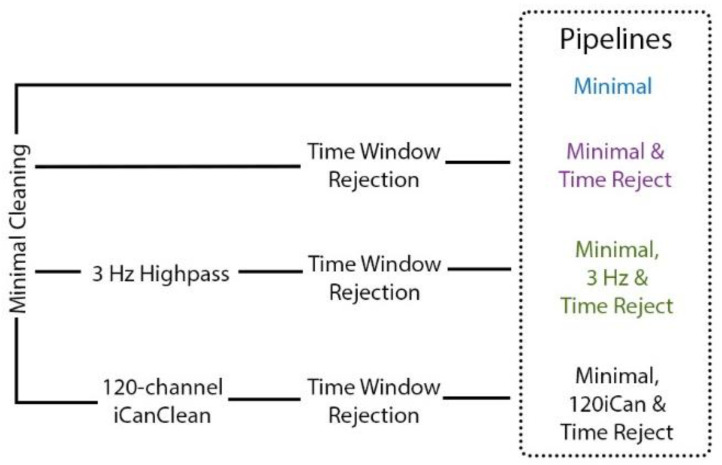
Overview of artifact removal pipelines. The Minimal pipeline (blue) includes basic preprocessing (filtering, channel rejection, interpolation, average re-referencing). The Minimal & Time Reject pipeline (purple) rejects bad time windows using clean_artifacts after minimal cleaning. The Minimal, 3 Hz & Time Reject pipeline (green) filters the scalp channels after minimal cleaning and with subsequent time window rejection (substituting for iCanClean). The Minimal, 120iCan & Time pipeline (black) uses all noise electrodes to clean scalp electrodes with canonical correlation analysis after minimal cleaning and with subsequent time window rejection.

**Figure 4 sensors-22-05867-f004:**
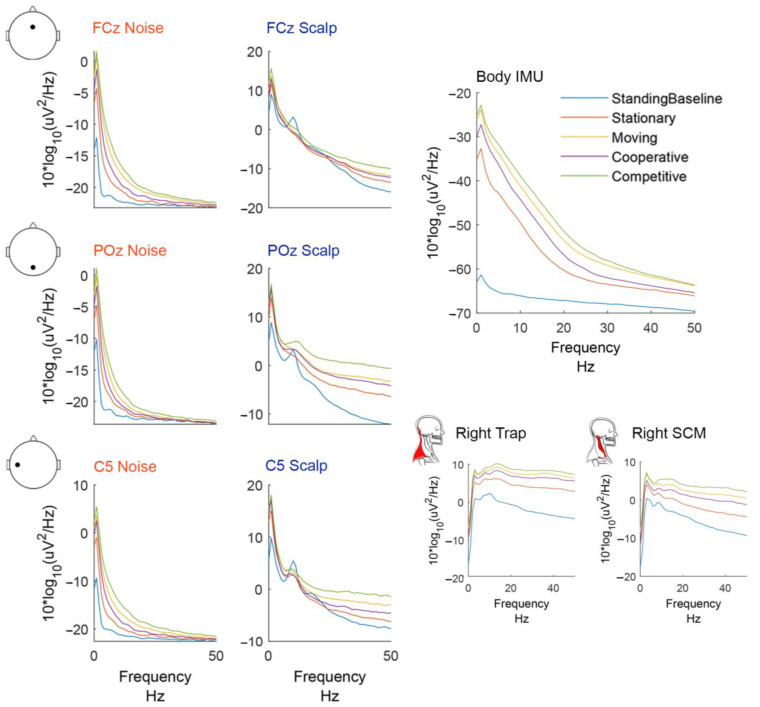
Group average (*n* = 20) power spectral density plots. Power spectral densities were computed using the EEGLAB spectopo function (pwelch, non-overlapping hamming windows, 250 window length) on epoched data −1 to 1 s around the participant hit events. Each condition is plotted in a different color.

**Figure 5 sensors-22-05867-f005:**
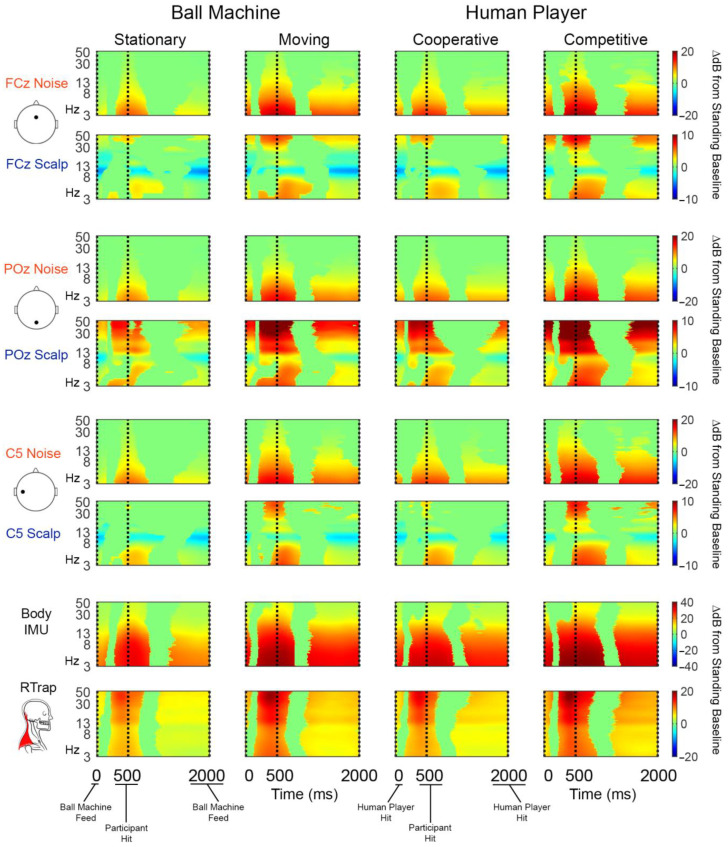
Group average (*n* = 20) event-related spectral perturbation (time-frequency) plots. Noise electrodes are in rows 1, 3, and 5. Scalp-matched electrodes are shown in rows 2, 4, and 6. The body IMU and right trapezius (RTrap) are shown in rows 7 and 8, respectively. Each hitting condition is shown in a different column. Significant increases in spectral power relative to standing baseline are in red and significant decreases in power relative to standing baseline are in blue. Vertical dashed lines indicate events in a single “swing cycle”. We defined a “swing cycle” as three events: (1) when the ball was presented to the participant as a ball machine feed or human player hit, (2) when the participant hit the ball, and (3) when the ball was presented to the participant for the next hit. Nonsignificant differences from baseline (bootstrap statistics with alpha 0.05) were masked to 0 dB (green).

**Figure 6 sensors-22-05867-f006:**
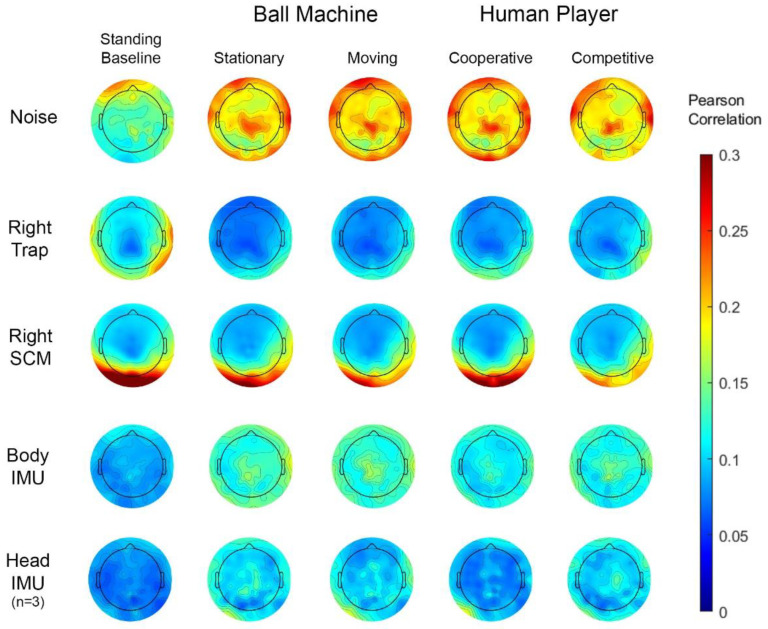
Group average time series correlations of individual scalp electrodes with paired noise electrodes (row 1), right trapezius (trap) muscle (row 2), right sternocleidomastoid muscles (row 3), and resultant body IMU acceleration (row 4) with *n* = 20 participants. The scalp electrode correlation with the resultant head IMU acceleration is shown in row 5 with *n* = 3 participants. Each column shows a different condition. Pearson’s R was converted to Fisher’s Z for all participants and all conditions, averaged together, and converted back to Pearson’s R.

**Figure 7 sensors-22-05867-f007:**
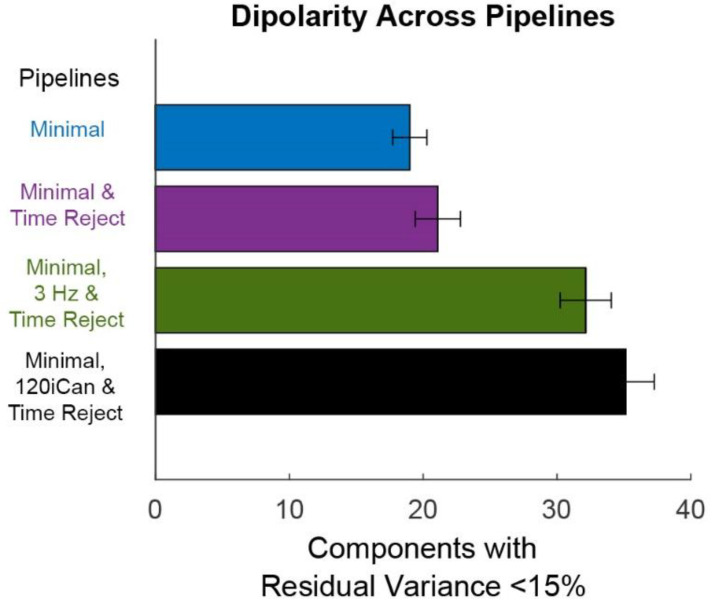
Mean +/− SEM of dipolarity shows the quality of the ICA decomposition. Dipolarity is measured as the number of components with residual variance less than 15% after fitting dipoles with a three-layer boundary element model of the standard MNI brain template. The Minimal pipeline (blue) includes basic preprocessing (filtering, channel rejection, interpolation, average re-referencing). The Minimal & Time Reject pipeline (purple) rejects bad time windows using clean_artifacts after minimal cleaning. The Minimal, 3 Hz & Time Reject pipeline (green) filters the scalp channels after minimal cleaning and with subsequent time window rejection (substituting for iCanClean). The Minimal, 120iCan, & Time Reject pipeline (black) uses all noise electrodes to clean scalp electrodes with canonical correlation analysis after minimal cleaning and with subsequent time window rejection.

**Figure 8 sensors-22-05867-f008:**
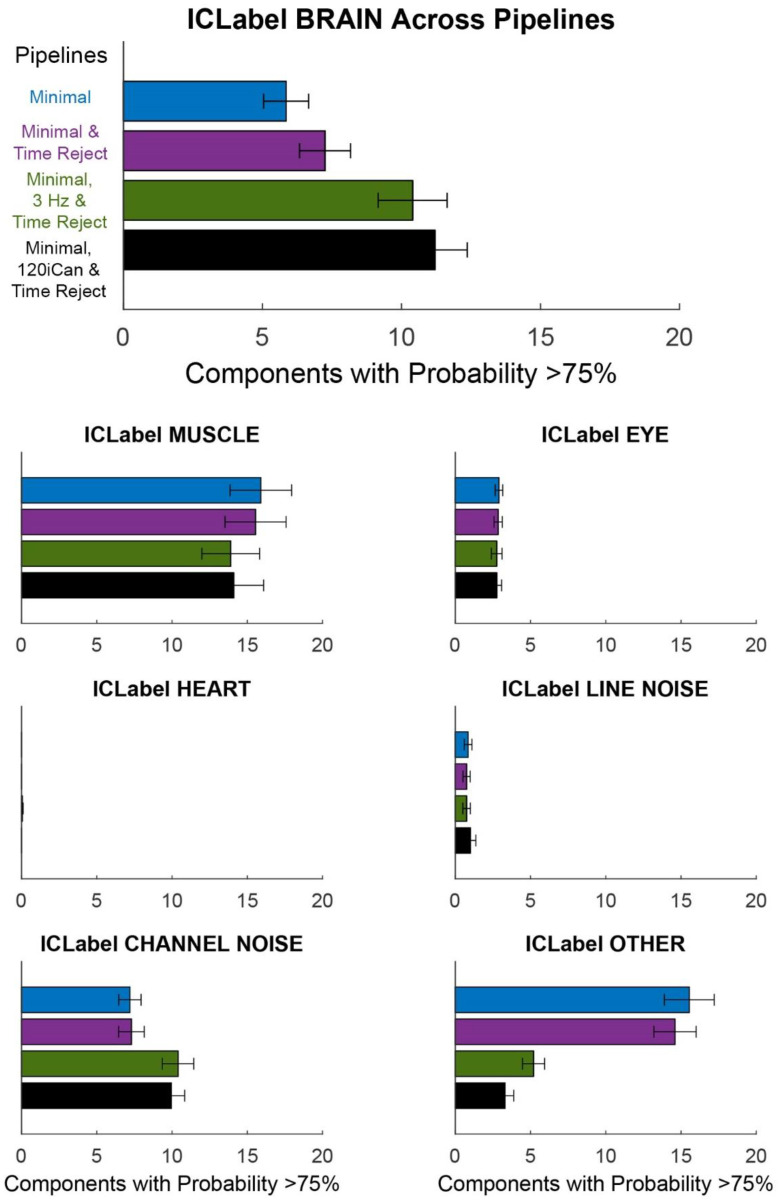
ICLabel classification of components for different pipelines. Each bar plot shows the mean +/− SEM of the number of components with classification probability >75% MUSCLE, EYE, HEART, LINE NOISE, CHANNEL NOISE, and OTHER. The Minimal pipeline (blue) includes basic preprocessing (filtering, channel rejection, interpolation, average re-referencing). The Minimal & Time Reject pipeline (purple) rejects bad time windows using clean_artifacts after minimal cleaning. The Minimal, 3 Hz, & Time Reject pipeline (green) filters the scalp channels after minimal cleaning and with subsequent time window rejection (substituting for iCanClean). The Minimal, 120iCan, & Time Reject pipeline (black) uses all noise electrodes to clean scalp electrodes with canonical correlation analysis after minimal cleaning and with subsequent time window rejection.

## Data Availability

Pipeline comparison data presented in this study are available in the Appendix A.

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
