# Peer review of "Characterizing and Removing Artifacts Using Dual-Layer EEG during Table Tennis"

_sensors, 2022, doi:10.3390/s22155867_

Round 1
Reviewer 1 Report
"Characterizing and Removing Artifacts using Dual-Layer EEG during Table Tennis" is an interesting article. The purpose of this study was to characterize and identify strategies to remove artifacts in EEG data for a discrete, responsive, whole-body task such as table tennis. The authors compared individual scalp channels, noise-matched channels, neck muscle channels and accelerometer data from the head and body to determine the efficacy of different reference signals in capturing noise. They compared artifact removal methods with and without the use of the dual-layer noise electrodes. The authors found that the dual-layer approach does well to characterize motion artifact affecting scalp channels during a dynamic sport like table tennis.
There are few issues in the article that need to be addressed.
Materials and Methods
Dual-Layer Approach
In line 124 it is suggested to add more information about amplifier characteristics, gain, low cut filters, high filters, the impedance of electrodes, sampling frequency, etc.
References
A careful review of each of the references is suggested.
Line 578. The volume of the magazine is 00 ?
Line 584. The volume and pages are repeated
Line 610. The name and additional data of the journal do not exist.
Line 623. The name and additional data of the journal do not exist.
Line 640. The volume of the magazine is 00 ?
Reviewer 2 Report
The paper is nicely written and the topic is interesting. I have some comments as follows.
(1) An objective performance (in numbers) can be reported in the abstract.
(2) Did the participants feel comfortable with the EEG headset during the match? If not, did it affect the result?
(3) Is there any comparison with other related methods?
Reviewer 3 Report
1. This study investigates and removes artifacts of dual-layer EEG for non-locomotor paradigms. Few healthy subjects are invited to play table tennis.
2. This study should concisely illustrate the artifacts removal by equations.
3. Time window rejection should be concisely formulated by equations.
4. Do all subjects reveal the same characteristics of artifacts?
5. This study should illustrate that why neural data in the delta and theta range (< 7 Hz) may have motion artifact contamination?
Round 2
Reviewer 3 Report
The authors have addressed my comments.